# Extracting regions of interest from biological images with convolutional sparse block coding

**Marius Pachitariu**[1], **Adam Packer**[2], **Noah Pettit**[2], **Henry Dagleish**[2],
**Michael Hausser**[2] **and Maneesh Sahani**[1]
[1]Gatsby Unit, UCL, UK {`marius, maneesh`}@gatsby.ucl.ac.uk
[2]The Wolfson Institute for Biomedical Research, UCL, UK {`a.packer,`
`noah.pettit.10, henry.dalgleish.09, m.hausser`}@ucl.ac.uk

## Abstract

Biological tissue is often composed of cells with similar morphologies replicated throughout large volumes and many biological applications rely on the accurate identification of these cells and their locations from image data. Here we develop a generative model that captures the regularities present in images composed of repeating elements of a few different types. Formally, the model can be described as convolutional sparse block coding. For inference we use a variant of convolutional matching pursuit adapted to block-based representations. We extend the K-SVD learning algorithm to subspaces by retaining several principal vectors from the SVD decomposition instead of just one. Good models with little cross-talk between subspaces can be obtained by learning the blocks incrementally. We perform extensive experiments on simulated images and the inference algorithm consistently recovers a large proportion of the cells with a small number of false positives. We fit the convolutional model to noisy GCaMP6 two-photon images of spiking neurons and to Nissl-stained slices of cortical tissue and show that it recovers cell body locations without supervision. The flexibility of the block-based representation is reflected in the variability of the recovered cell shapes.

## 1   Introduction

For evolutionary reasons, biological tissue at all spatial scales is composed of repeating patterns. This is because successful biological motifs are reused and multiplied by evolutionary pressures. At a small spatial scale eukaryotic cells contain only a few types of major organelles like mitochondria and vacuoles and several dozen minor organelles like vesicles and ribosomes. Each of the organelles is replicated a large number of times within each cell and has a distinctive visual appearance. At the scale of whole cells, most tissue types like muscle and epithelium are composed primarily of single cell types. Some of the more diverse biological tissues are probably in the brain where gray matter contains different types of neurons and glia, often spatially overlapping. Repetition is also encouraged at large spatial scales. Striate muscles are made out of similar axially-aligned fibers called sarcomers and human cortical surfaces are highly folded inside the skull producing repeating surface patterns called gyri and sulci.

Much biological data at all spatial scales comes in the form of two- or three-dimensional images. Non-invasive techniques like magnetic resonance imaging allow visualization of details on the order of one millimeter. Cells in tissue can be seen with light microscopy and cellular organelles can be seen with the electron microscope. Given the stereotypical nature of biological motifs, these images often appear as collections of similar elements over a noisy background, as shown in figure 1(a). We developed a generative image model that automatically discovers the repeating motifs, and segments biological images into the most common elements that form them. We apply the model to two-dimensional images composed of several hundred cells of possibly different types, such as

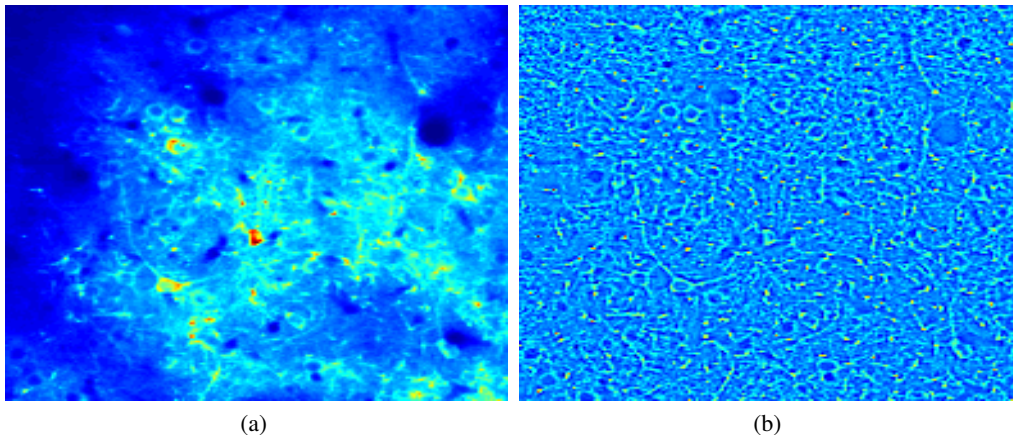

|      |      |
|:----:|:----:|
| (a)  | (b)  |

Figure 1: **a.** Mean image of a two-photon recording of calcium-based fluorescence. **b.** Same image as in (a) after subtractive and divisive normalization locally.

images of cortical tissue expressing fluorescent GCaMP6, a calcium indicator, taken with a two-photon microscope in vivo. We also apply the model to Nissl-stained cortical tissue imaged in slice. Each experimental exposure can contain hundreds of cells and many exposures are usually taken over a single experimental session. Our main aim is to automate the cell detection stage, because tracing cell contours by hand can be a laborious and inexact process, especially given the multitude of confounds usually present in these images. One confound clearly visible in figure 1(a) is the large variation in contrast and luminance over a single image. A second confound, also visible in figure 1(a), is that many cells tend to cluster together and press their boundaries against each other. Assigning pixels to the correct cell can be difficult. A third confound is that calcium, the marker which the fluorescent images report, is present in the entire neuropil (in the dendrites and axons of the cells). Activation of calcium in the neuropil makes a noisy background for the estimation of cell somata. Given such large confounds, a properly-formulated image model is needed to resolve the ambiguities as well as the human eye can resolve them.

## 1.1 Background on automated extraction of cell somata

Histological examination of biological tissue with light-microscopy is an important application for techniques of cell identification and segmentation. Most algorithms for identifying cell somata from such images are based on hand-crafted filtering and thresholding techniques. For example, [1] proposes a pipeline of as many as fourteen separate steps, each of which is meant to deal with some particular dimension of variability in the images. Our approach is to instead propose a fully generative model of the biological tissue which encapsulates our beliefs about the stereotypical structure of such images. Inference in the model inverts the generative model — or in other words deconvolves the image — and thereby replaces the filtering and thresholding techniques usually employed. Learning the parameters of the generative model replaces the hand-crafting of the filters and thresholds.

For one image type we use here, fluorescent images of neuronal tissue, the approach of [2] is closer in spirit to our methodology of model design and inference. The authors propose an independent components analysis (ICA) model of the movies which expresses their beliefs that all the pixels belonging to a cell should brighten together, but only rarely. The model effectively uses the temporal correlations between pixels to segment each image, much like [3] but the pipeline of [3] is manual and not model-designed like that of [2]. Both of these studies are different from our approach, because we aim to recover cell bodies from single images alone. The method of [2] applies well to small fields of view and large coherent fluorescence fluctuations in single cells, but fails when applied to our data with large fields of view containing hundreds of small neurons. The failure is due to long-range spatial correlations between many thousands of pixels which overcome the noisy correlations between the few dozen pixels belonging to each cell. Consequently, the independent components extracted by the algorithm of [2][1] have large spatial domains as can be seen in supplemental figure 1. Our approach is robust to large non-local correlations because we analyze the

mean image alone. One advantage is that the resulting model can be applied not just to data from functional imaging experiments but to data from any imaging technique.

## 1.2 Background on convolutional image models

Our proposed image model is a novel extension of a family of recent algorithms based on sparse coding that are commonly used in object recognition experiments [4], [5], [6], [7], [8]. A starting point for our model was the convolutional matching pursuit (MP) implementation of [5] (but see [6] for more details). The authors show that convolutional MP learns a diverse set of basis functions from natural images. Most of these basis functions are edges, but some have a globular appearance and others represent curved edges and corners. Their implied generative model of an image is to pick out randomly a few basis functions and place them at random locations. While this is a poor generative model for natural images, it is much better suited to biological images which are composed of many repeating and seemingly randomly distributed elements of a few different types.

One disadvantage of convolutional MP as described by [6] is that it uses fixed templates for each dictionary element. Although it seems like the cells in figure 1(b) might be well described by a single ring shape, there are size and shape variations which could be better captured by more flexible templates. In general, we expect the repeating elements in a biological image to have similar appearances to a first approximation, but patterned variability is unavoidable. A better model of the image of a single cell might be to assume it was generated by combining a few different prototypes with different coefficients, effectively interpolating between the prototypes. We group the prototypes related to a single object into blocks and every image is formed by activating a small number of such blocks. We call this model sparse block coding. Note that the blocking principle is common in natural image modelling, where Gabor filters in quadrature are combined with different coefficients to produce edges of different spatial phases. Independent subspace analysis (ISA [7]) also entails distributing basis functions into non-overlapping blocks. However, in our formulation the blocks are either activated or not, while ISA assumes a continuous distribution on the activations of each block. This property of sparse block coding makes it valuable in making hard assignments of inferred cell locations, rather than giving a continuous coefficient for each location.

Closer to our formulation, [8] have used a similar sparse block coding model on natural movie patches and added a temporal smoothness prior on the activation probabilities of blocks in consecutive movie frames. The expensive variational iterative techniques used by [8] for inference and learning in small image patches are computationally infeasible for the convolutional model of large images we present here. Instead, we use a convolutional block pursuit technique which is an extension of standard matching pursuit and has similarly low computational complexity even for arbitrarily large blocks and arbitrarily large images.

# 2 Model

## 2.1 Convolutional sparse block coding

Following [8], we distinguish between identity and attribute variables in the generative model of each object in an image. An object can be a cell, a cell fragment or any other spatially-localized object. Identity variables $\mathbf{h}_{xy}^k$, where $(x, y)$ is the location of the object and $k$ the type of object, are Bernoulli-distributed with very small prior probabilities. Each of the objects also has several continuous-valued attribute variables $\mathbf{x}_{xy}^{kl}$, with $l$ indexing the attribute. In the generative model these attributes are given a broad uniform probability and specify the coefficients with which a set of basis functions $A_{kl}$ are combined at spatial location $(x, y)$ before being linearly combined with objects generated at other locations. The full description of the generative process is best captured in terms of two-dimensional convolutions by the following set of equations

$$\mathbf{h}_{xy}^k \sim \text{Bernoulli}(p)$$
$$\mathbf{x}_{xy}^{kl} \sim \mathcal{N}\left(0, \sigma_x^2\right)$$
$$\mathbf{y} \sim \sum_{k,l} A_{kl} * \left(\mathbf{x}^{kl} \circ \mathbf{h}^k\right) + \mathcal{N}\left(0, \sigma_y\right),$$

where $\sigma_y$ is the (small) noise variance for the image, $\sigma_x$ is the (large) prior variance for the co-efficients, $p$ is a small activation probability specific to each object type, $\mathbf{h}^k$ and $\mathbf{x}^{kl}$ represent the full two-dimensional maps of the binary and continuous coefficients respectively, "$\circ$" represents the elementwise or Hadamard product and "$*$" denotes two-dimensional convolution where the result is

taken to have the same dimensions as the input image.[2] The joint log-likelihood (or negative energy) can now be derived easily

$$\mathcal{L}\left(\mathbf{x}, \mathbf{h}, A\right) = -\frac{\|\mathbf{y} - \sum_{k,l} A_{kl} * \left(\mathbf{x}^{kl} \circ \mathbf{h}^{k}\right)\|^2}{2\sigma_y^2} - \frac{\sum_{klxy}\left(\mathbf{x}_{xy}^{kl}\right)^2}{2\sigma_x^2} +$$

$$\sum_{kxy}\left(\mathbf{h}_{xy}^{k}\log(p) + (1 - \mathbf{h}_{xy}^{k})\log(1-p)\right) + \text{constants} \qquad (1)$$

In practice, we used $\sigma_x = \infty$ as we found that it gave similar results to finite values of $\sigma_x$. This model can be fit by alternately optimizing the cost function in equation 1 over the unobserved variables $\mathbf{x}$ and $\mathbf{h}$ and the parameters $A$. The prior bias parameter $p$ will not be optimized over but instead will be adjusted so as to guarantee a mean number of elements per image. We also set $\|A^{kl}\| = 1$ without loss of generality, since the absolute values of $\mathbf{x}$ can scale to compensate.

## 2.2 Inference by convolutional block pursuit

Given a set of basis functions $A_{kl}$ and an image $\mathbf{y}$, we would like to infer the most likely locations of objects of each type in an image. This inference is generally NP-hard but good solutions can nonetheless be obtained with greedy methods like matching pursuit (MP). In standard matching pursuit, a sequential process is followed where at each step a basis function $A_{kl}$ is chosen which if activated increases most the log-likelihood of equation 1. In our model, at each step we activate a full block $k$ which includes multiple templates $A^{kl}$. Due to the quadratic nature of equation 1, for a proposal $\mathbf{h}_{xy}^{k} = 1$ we can easily compute the MAP estimate for each $\mathbf{x}_{xy}^{k}$ given the current residual image $\mathbf{y}_{\text{res}} = \mathbf{y} - \sum_{k,l} A_{kl} * \left(\mathbf{x}^{kl} \circ \mathbf{h}^{k}\right)$. Here we understand $\mathbf{x}_{xy}^{k}$ as a vector concatenating $\mathbf{x}_{xy}^{kl}$ for all $l$. The MAP estimate for $\mathbf{x}_{xy}^{k}$ is

$$\hat{\mathbf{x}}_{xy}^{k} = \left((A^k)^T A^k\right)^{-1} \mathbf{v}_{xy}^{k}$$

$$\mathbf{v}_{xy}^{k}(l) = \left(\bar{A}^{kl} * \mathbf{y}_{\text{res}}\right)_{xy}$$

where $\bar{A}^{kl}$ is the basis function $A^{kl}$ rotated by 180 degrees and the matrix $A^k$ contains as columns the vectorized basis functions $A^{kl}$. The corresponding increase in likelihood in equation 1 is

$$\delta\mathcal{L}_{xy}^{k} = \frac{\left(\mathbf{v}_{xy}^{k}\right)^T \hat{\mathbf{x}}_{xy}^{kl}}{2\sigma_y^2} - \log\frac{p}{1-p}.$$

Inference stops when the activation penalty $\log\dfrac{p}{1-p}$ from the prior overcomes the data term for all possible objects $k$ at all possible locations $(x, y)$.

A simple trick common to all matching pursuit algorithms [9], [6] allows us to save computation when sequentially calculating $\mathbf{v}_{klxy} = \bar{A}^{kl} * \mathbf{y}_{\text{res}}$ by keeping track of $\mathbf{v}$ and updating it after each new coefficient is turned on:

$$\mathbf{v}^{\text{new}} = \mathbf{v} - \mathbf{G}_{(....),(k.xy)}\hat{\mathbf{x}}_{xy}^{k},$$

where $\mathbf{G}$ is the grand Gram matrix of all basis functions $A_{xy}^{kl}$ at all positions $(x, y)$, and the indexing means that every dot runs over all possible values of that index. Because the basis functions are much smaller in length and width than the entire image, most entries in the Gram matrix are actually 0. In practice, we do not keep track of these and instead keep track only of $\mathbf{G}_{(k'l'x'y'),(klxy)}$ for $|x - x'| < d$ and $|y - y'| < d$, where $d$ is the width and length of the basis function. We also keep track during inference of $\hat{\mathbf{x}}$ and $\delta\mathcal{L}_{xy}^{k}$ and only need to update these quantities at positions $(x, y)$ around the extracted object. These caching techniques make the complexity of the inference scale linearly with the number of objects in each image, regardless of image or object size.

Thus, our algorithm benefits from the computational efficacy of matching pursuit. One additional computation lies in determining the inverse of $(A^k)^T A^k$ for each $k$. This cost is negligible, since each block contains a small number of attributes and we only need to do the inversions once per iteration. Every iteration of block pursuit requires updating $\mathbf{v}$, $\hat{\mathbf{x}}$ and $\delta\mathcal{L}_{xy}^{k}$ locally around the extracted

block, which is several times more expensive than the corresponding update in simple matching pursuit. However, this cost is also negligible compared to the cost of finding the best block at each iteration: the single most intensive operation during inference is the loop through all the elements in all the convolutional maps to find the block which most increases the likelihood if activated. All the other update operations are local around the extracted block, and thus negligible. In practice for the datasets we use (for example, 18 images of 256 by 256 pixels each), a model can be learned in minutes on a modern CPU and inference on a single large image takes under one second.

## 2.3 Learning with block K-SVD

Given the inferred active blocks and their coefficients, we would like to adapt the parameters of the basis functions $A^{kl}$ so as to maximize the cost function in eq 1. This can most easily be accomplished by gradient descent (GD). Unfortunately, for general dictionary learning setups gradient descent can produce suboptimal solutions, where a proportion of the basis function fail to learn meaningful structure [10]. Similarly, for our block-based representations we found that gradient descent often mixed together subspaces that should have been separated (see fig 2(c)). We considered the option of estimating the subspaces in each $A^k$ sequentially where we run a couple of iterations of learning with a single subspace in each $A^k$ and then every couple of iterations we increase the number of subspaces we estimate for $A^k$. This incremental approach always resulted in demixed subspaces like those in figure 2(a). Note also that the standard approach in MP-based models is to extract a fixed number of coefficients per image, but in our database of biological images there are large variations in the number of cells present in each image so we needed the inference method to be flexible enough to accomodate varying numbers of objects. To control the total number of active coefficients, we adjusted during learning the prior activation probability $p$ whenever the average number of active elements was too small or too large compared to our target mean activation rate.

Although incremental gradient descent worked well, it tended to be slow in practice. A popular learning algorithm that was proposed to accelerate patch-based dictionary learning is K-SVD [10]. In every iteration of K-SVD, coefficients are extracted for all the image patches in the training set. Then the algorithm modifies each basis function sequentially to exactly minimize the squared reconstruction cost. The convolutional MP implementation of [6] indeed uses K-SVD for learning and we here show how K-SVD can be adapted to block-based representations.

At every iteration of K-SVD, given a set of active basis functions per image obtained with an inference method, the objective is to minimize the reconstruction cost with respect to the basis functions and coefficients simultaneously [10]. We consider each basis function $A^{kl}$ sequentially, extract all image patches $\{y_i\}_i$ where that basis function is active and assume all coefficients for the other basis functions are fixed. In the convolutional setting, these patches are extracted from locations in the images where each basis function is active [6]. We add back the contribution of basis function $A^{kl}$ to each patch in $\{y_i\}_i$ and now make the observation that to minimize the reconstruction error with a single basis function $\hat{A}^{kl}$ we must find the direction in pixel space where most of the variance in $\{y_i\}_i$ lies. This can be done with an SVD decomposition followed by retaining the first principal vector $\hat{A}^{kl}$. The new reconstructions for each patch $y_i$ are $y_i - \hat{A}^{kl}(\hat{A}^{kl})^T y_i$ and with this new residual we move on to the next basis function to be reestimated.

By analogy, in block K-SVD we are given a set of active blocks per image, each block consisting of $K$ basis functions. We consider each block $A^k$ sequentially, extract all image patches $\{y_i\}_i$ where that block is active and assume all coefficients for the other blocks are fixed. We add back the contribution of block $A^k$ to each patch in $\{y_i\}_i$ and like before perform an SVD decomposition of these residuals. However, we are now looking for a $K$-dimensional subspace where most of the variance in $\{y_i\}_i$ lies and this is exactly achieved by considering the first $K$ principal vectors returned by SVD. The reconstructions for each patch are $y_i - \hat{A}^k(\hat{A}^k)^T y_i$ where $\hat{A}^k$ are the first $K$ principal vectors. On a more technical note, after each iteration of K-SVD we centered the parameters spatially so that the center of mass of the first direction of variability in each block was aligned to the center of its window, otherwise the basis functions did not center by themselves.

Although K-SVD was an order of magnitude faster than GD and converged in practice, we noted that in the convolutional setting K-SVD is biased. This is because at the step of re-estimating a block $A^k$ from a set of patches $\{y_i\}_i$, some of these patches may be spatially overlapping in the full image. Therefore, the subspaces in $A^k$ are driven to explain the residual at some pixels multiple times. One way around the problem would be to enforce non-overlapping windows during inference,

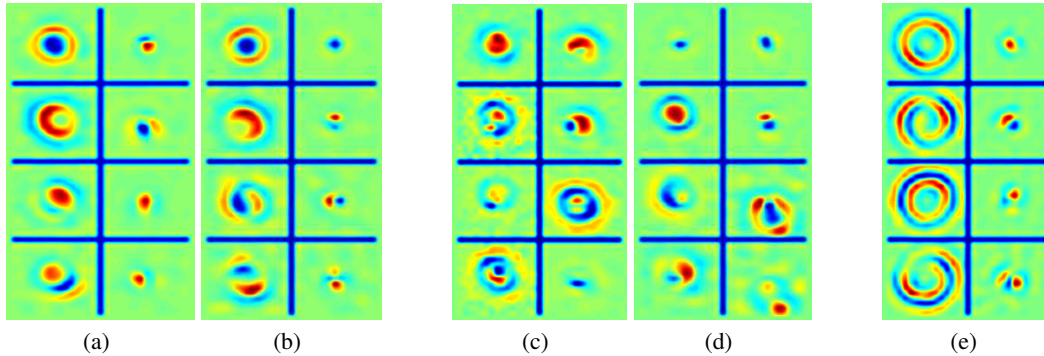

| (a) | (b) | | (c) | (d) | | (e) |

Figure 2: **a.** Typical recovered parameters with incremental gradient descent learning on GCaMP6 fluorescent images. Each column is a block and is sorted in the order of variance from the SVD decomposition. Left columns capture the structure of cell somatas, while right columns represent dendrite fragments. **b.** Like (*a*) but with incremental block K-SVD. Similar subspaces are recovered with ten times fewer iterations. **c.** and **d.** Typical failure modes of learning with non-incremental gradient descent and block K-SVD, respectively. The subspaces from (*a*) appear mixed together. **e.** Subspaces obtained from Nissl-stained slices of cortex.

but in our images many cell pairs touch and would in fact require overlapping windows. Instead, we decided to fine-tune the parameters returned by block K-SVD with a few iterations of gradient descent which worked well in practice and in simulations recovered good model parameters with little further computational effort.

## 3 Results

### 3.1 Qualitative results on fluorescent images of neurons

The main applications of our work are to nissl-stained slices and to fields of neurons and neuropil imaged with a two-photon microscope (figure 1(a)). The neurons were densely labeled with a fluorescent calcium indicator GCaMP6 in a small area of the mouse somatosensory (barrel) cortex. While the mice were either anesthetized or awake, their whiskers were stimulated which activated corresponding barrel cortex neurons, leading to an influx of calcium into the cells and consequently an increase in fluorescence which was reported by the two-photon microscope. Although cell somas receive a large influx of calcium, dendrites and axons can also be seen. Individual images of the fluorescence can be very noisy purely due to the low number of photons released over each exposure. Better spatial accuracy can be obtained at the expense of temporal accuracy or at the expense of a smaller field of view. In practice, cell locations can be identified based on the mean images recorded over the duration of an entire experiment, in our case 1000 or 5000 frames. Using 18 images like the one in figure 1(b) we learned a full model with two types of objects each with three subspaces. One of the object types, the left column in figure 2(a) was clearly a model of single neurons. The right column of figure 2(a) represented small pieces of dendrite that were also highly fluorescent. Note how within a block each of the two objects includes dimensions of variability that capture anisotropies in the shape of the cell or dendritic fragments. Figure 3(a) shows in alternating odd rows patches from the training set identified by the algorithm to contain cells and the respective reconstructions in the even rows. Note that while most cells are ring-shaped, some appear filled and some appear to be larger and the model's flexibility is sufficient to capture these variations. Figure 2(c) shows a typical failure for gradient based learning that motivated us to use incremental block learning. The two subspaces recovered in figure 2(a) are mixed in figure 2(c) and the likelihood from equation 1 is correspondingly lower.

### 3.2 Simulated data

We ran extensive experiments on simulated data to assess the algorithm's ability to learn and infer cell locations. There are two possible failure modes: the inference algorithm might not be accurate enough or the learning algorithm might not recover good parameters. We address each of these failure modes separately. We wanted to have simulated data as similar as possible to the real data so we first fitted a model to the GCaMP6 data. We then took the learned model and generated a new dataset from it using the same number of objects of each type and similar amounts of Gaussian noise as the real images. To generate diverse shapes of cells, we fit a $K$-dimensional multivariate Gaussian

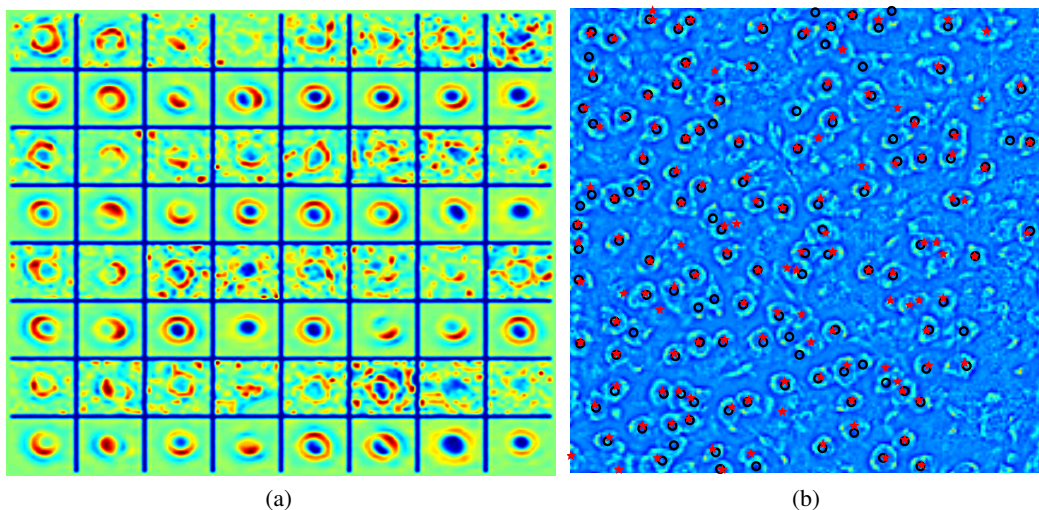

(a)                                                  (b)

Figure 3: **a.** Patches from the GCaMP6 training images (odd rows) and their reconstructions (even rows) with the subspaces shown in figure 2(b). **b.** One area from a Nissl-stained image together with a human segmentation (open circles) and the model segmentation (stars). Larger zoom versions are available in the supplementary material.

to the posteriors of each block on the real data and generated coefficients from this model for the simulated images. Supplemental figure 6 shows a simulated image and it can be seen to resemble images in the training set. Note that we are not modelling some of the structured variability in the noise, for example the blood vessels and dendrites visible in figure 1(b). This structured variability is the likely reason why the model performs better on simulated than on real images.

### 3.2.1   Inference quality of convolutional block pursuit

We kept the ground truths for the simulated dataset and investigated how well we can recover cell locations when we know perfectly what the simulation parameters were. There is one free parameter in our model that we cannot learn automatically which is the average number of extracted objects per image. We varied this parameter and report ROC curves for true positives and false positives as we vary the number of extracted coefficients. Sometimes we observed that cells were identified not exactly at the correct location but one or a few pixels away. Such small deviations are acceptable in practice, so we considered inferred cells as correctly identified if they were within four pixels of the correct location (cells were 8-16 pixels in diameter). We enforced that a true cell could only be identified once. If the algorithm made two predictions within ¡4 pixels of a true cell, only the first of these was considered a true positive. Figure 4(a) reports the typical performance of convolutional block pursuit. We also investigated the quality of inference without considering the full structure of the subspaces in each object. Using a single subspace per object is equivalent to matching pursuit, achieved significantly worse performance and saturated at a smaller number of true positives because the model could not recognize some of the variations in cell shape.

### 3.2.2   Learning quality of K-SVD + gradient descent

We next tested how well the algorithm recovers the generative parameters. We assume that the model knows how many object types there are and how many attributes each object type has. To compare the various learning strategies we could in principle just evaluate the joint log-likelihood of equation 1. However the differences, although consistent, were relatively small and hard to interpret. More relevant to us is the ROC performance in recovering correctly cell locations. Block K-SVD consistently recovers good parameters but does not perform quite as well as the true parameters because of its bias (figure 4(b)). However refinement with GD consistently recovers the best parameters which approach the performance of the true generative parameters. We also asked how well the model recovers the parameters when the true number of objects per image is unknown, by running several experiments with different mean numbers of objects per image. The performance of the learned subspaces is reported in figure 4(c). Although the correct number of elements per image was 600, learning with as few as 200 or as many as 1400 objects resulted in equally well-performing models. If performance on simulated data is at all indicative of behavior on real data, we conclude that our algorithm is not sensitive to the only free parameter in the model.

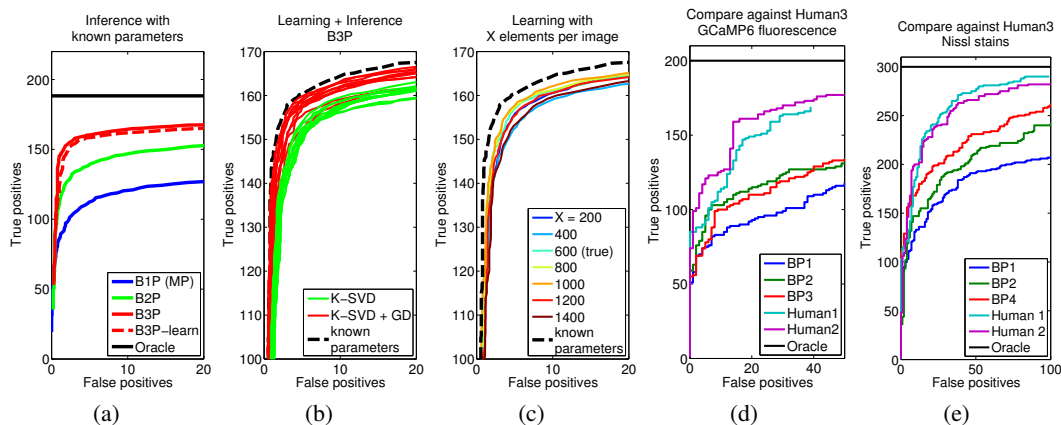

Figure 4: ROC curves show the model's behavior on simulated data (a-c) and on manually-segmented GCaMP6 images (d) and Nissl-stained images (e) . **a.** Inference with block pursuit with all three subspaces per object (B3P) as well as block pursuit with only the first or first two principal subspaces (B1P and B2P). We also show for comparison the performance of B3P with model parameters identified by learning. Notice the small number of false negatives when a large proportion of the cells are identified. The cells not identified were too dim to pick out even with a large number of false negative, hence the quick saturation of the ROC curve. **b.** Ten runs of block K-SVD followed by gradient descent. Refining with GD improved performance. **c.** Not knowing the average number of elements per image does not make a difference on simulated data.

## 3.3 Comparison with human segmentation on biological images

We compare the segmentation of the model with manual segmentations on one example each of the GCaMP6 and Nissl-stained images (figures 4(d) and 4(e)). The human segmenters were instructed to locate cells in approximately the order of confidence, thus producing an ordering similar to the ordering returned by the algorithm. As we retain more cells from that ordering we can build ROC curves showing the agreement of the humans with each other, and of the model's segmentation to the humans'. We found that using multiple templates per block helped the model agree more with the human segmentations. In the case of the Nissl-stain, block coding with four templates identified fifty more cells than matching pursuit. Although the model generally performs below inter-human agreement, the gap is sufficiently small to warrant practical use. In addition, a post-hoc analysis suggests that many of the model's false positives are in fact cells that were not selected in the manual segmentations. Examples of these false positives can be seen both in figure 3(b) and in figures in the supplementary material. As we anticipated in the introduction, a standard method based on thresholded and localized correlation maps only reached 25 true positives at 50 false positives and is not shown in figure 4(d).

## 4 Conclusions

We have presented an image model that can be used to automatically and effectively infer the locations and shapes of cells from biological image data. This application of generative image models is to our knowledge novel and should allow automating many types of biological studies. Our contribution to the image modelling literature is to extend the sparse block coding model presented in [8] to the convolutional setting where each block is allowed to be present at any location in an image. We also derived convolutional block pursuit, a greedy inference algorithm which scales gracefully to images of large dimensions with many possible object types in the generative model. For learning the model, we extended the K-SVD learning algorithm to the block-based and convolutional representation. We identified a bias in convolutional K-SVD and used gradient descent to fine-tune the model parameters towards good local optima.

On simulated data, convolutional block pursuit recovers with good accuracy cell locations in simulated biological images and the learning rule recovers well and consistently the parameters of the generative model. Using the block pursuit algorithm recovers significantly more cells than simple matching pursuit. On data from calcium imaging experiments and nissl-stained tissue, the model succeeds in recovering cell locations and learns good models of the variability among different cell shapes.

## Footnotes

[1] available online at http://www.snl.salk.edu/∼emukamel/

[2]In other words, the convolution uses "zero-padding".

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
