[Supplementary Material]



Figure 5: Failure of a correlation-based segmentation technique (independent components analysis) to segment our GCaMP6 fluorescence movies. Shown is one typical independent component. We believe the failure is due to global correlations present over large areas of the cortex which overwhelm the local correlations due to pixels belonging to the same cells.

Figure 6: Like figure 4(d) in the main text, this shows the extent to which the automated methods agree with the human segmentations. We added in this figure the performance of an automated method based on correlations within the entire movies (autoc), to show its poor performance on our data.

Figure 7: Probabilistic sample simulated from a model fit to the GCaMP6 average fluorescence data set. A dataset of 20 such images was used to evaluate the performance of the algorithm.

Figure 8: Larger field of view for the same imaging session as in figure 3b.

Figure 9: Detail of the segmentation on the Nissl-stained tissue.

Figure 10: Segmentation of GCaMP6 mean fluorescence overlay on the contrast normalized image (see figure 1).