[Reviews · NeurIPS 2013]

Submitted by Assigned_Reviewer_2

Summary: the authors developed a novel method specifically to address an increasingly important problem in neuroscience and cell biology more generally: extract cell bodies from noisy images. they adapt a well-known dictionary learning method, k-svd, to this domain, and generalize the inference scheme to make computations for efficient. The images demonstrate fruitful results, and the quantitative results demonstrate useful performance, although not at the level of a human expert.

Quality: i very much like the paper. goal & methods were defined clearly, performance is useful. some suggestions for possible improvements/extensions: the method does not take into account time-varying information. when 2 cells are spatially overlapping (eg, in different planes), then the only information enabling separation might be time-varying information. moreover, if there is drift in the image, potentially due to animal movement, then averaging across all the images will suffer, and some additional tracking of cells might be useful. also, it is relatively common for experimentalists to collect multiple channels, say one structural and one functional. this method could utilize that information to help improve SNR. It would have been nice to see a quantitative comparison between this method and any other previously proposed method. many others have proposed methods to extract shapes, and specifically cells, from images. i imagine this method would outperform the others, at least in terms of computational complexity for some of them, but maybe not.

Clarity: i found the paper to be clear.

Originality: a few modifications of k-svd, and a useful and interesting application, is sufficiently original for me.

Significance: if the code is shared, i imagine this work could be highly significant. otherwise, i imagine the authors will have to write another paper targeted at the neuroscience/biology community to express the ideas in a language they are more comfortable with.
Summary: i believe this manuscript makes a useful contribution to the nips community, by extending dictionary learning methods and applying them fruitfully to an interesting and important application. the paper could be improved, imho, by comparing with other methods and sharing the code.

Submitted by Assigned_Reviewer_4

Article summary:
This paper presents a generative model to infer cell shape and location in biological images. This is a different approach than is often found in this field, where filter-tuning and adaptive thresholding are common.

A convolutional sparse block coding model is used, together with a matching pursuit inference step. The random spatial repetition of numerous, yet similar, motifs within their image targets is claimed to be well suited for a convolutional matching pursuit approach. In order to provide flexibility in the definition of their target motifs, they employ sparse block coding - a technique similar to using a weighted subset of Gabor patches. In their implementation, they only use binary weighting to ease subsequent assignment thresholding.

To determine the set of basis functions, a block based K-SVD algorithm, with an additional gradient descent step, was designed and shown to perform well.

The authors show results on simulated and real-world data, and report good performance.

Quality:
I have difficulty judging the quality of this work as it is outside my field. The paper appears moderately well written, though a more scientific phrasing would be appreciated.

Minor points:
The title of reference (3) is incorrect.

Clarity:
I found the paper moderately clear, though I cannot say that I fully appreciated all the steps within their method.

Figure 3: It is difficult to judge when the red stars and black circles are co-located, especially at the image size used. Alternative labelling, or a larger/zoomed image would allow the reader to better assess the results.

Originality:
I am unable to assess this - outside of my field.

Significance:
Given the information presented in the paper, the significance to the biological imaging community could be high, as the process is automatic and seems robust to parameter variation.

Rebuttal:
I have read the author rebuttal and other reviews. I will keep my original score.
Summary: The paper presents the application of techniques that are used in other imaging fields to that of biological imaging for the purpose of cell-centre delineation. The algorithms are adapted to both the data sources and the inherent image confounds, they demonstrate how the training of the method is achieved, and finally report good performance.

Submitted by Assigned_Reviewer_5

The authors propose a generative model for biological images composed of repeating elements of a small number of classes (we could also call these "textures"). The model is formalized with a convolutional sparse block coding structure. The authors propose a credible learning algorithm, and perform a battery of validation experiments for simulated and real biological imaging data.

This paper is relatively clear and of high quality. It takes well-defined methodology, extends it, and applies it to a worthy application. I appreciate the full explanation of the results with their method with one slightly less sophisticated (Figure 2).

The work appears to be fairly original in it's application, though it uses well-known techniques. It is unclear what impact it will have in the machine learning community, but it has the potential to be very useful to biological scientists.
Summary: A thorough paper with a good potential impact.
Author Feedback

Author rebuttal: We thank the reviewers for their comments and helpful suggestions. While all three reviewers seem to have been generally positive, they do raise some points which we feel it appropriate to address.

First, all three reviewers characterise the algorithmic contributions as straightforward extensions of standard image processing methods [For example, Rev_5 states that the paper uses "well-known" machine learning techniques and is unsure of the potential impact in the machine learning community.] While the general framework of generative model design, inference and learning are well established, with matching pursuit (MP) and K-SVD both very widely exploited for their computational efficiency, our extension of these algorithms to the “block” setting of subspace templates is novel and non-trivial, and may well find broad applicability in general image processing.

Standard MP efficiently decomposes a signal into sparsely-distributed components whose appearances match a fixed (generally small) set of templates. Block MP extends this approach to sparse components with variable appearance, while retaining the computational efficiency. In combination with the extended learning algorithm provided by the novel block K-SVD algorithm [it is worth noting that the “K” in the original K-SVD may be misleading and in fact standard K-SVD only retains the single direction of maximal variance as the shape of each filter], we thus have what we believe is the first efficient, scalable approach to sparse subspace-based decomposition.

Matching pursuit and K-SVD are some of the most widely-used practical machine learning tools, with the original papers amassing 5300 and 1600 citations respectively. However, the subspace-based sparse priors of block MP and block K-SVD are likely to provide a better description of many different sorts of signal than the standard sparse-template prior, while retaining the efficiency of the basic methods. Thus, we expect these new methods to find application beyond the micrographic segmentation problems considered here, in the analysis of the broad range of signals with which sparse coding is used: including other images, sounds, radar, remote-sensing etc. Although we did not have room to consider these other data types here, we did show successful results on different classes of micrographic image, with different underlying signals and structures.

A third algorithmic development is of a more technical nature: extending the algorithms to deal with full-sized images required careful book-keeping of image reconstruction, active blocks and caching of the filtered convolutional maps. Almost all image models in the literature are applied to small image patches (due to high computational complexity), with a few notable exceptions like [6].

Rev_2 suggested that we add comparisons to previous cell detection algorithms. We did want very much to include such a comparison but could not find published code for fully automated systems; instead available software packages seemed to be designed to help the segmentation process for the human annotator. We did compare performance to our own implementation of a more standard correlation-based segmentation algorithm, but this performed much more poorly than our new algorithm (see figure 6 of the supplementary material). The qualitative failure of activity-based ICA (when applied on large fields of view) is also shown in supplemental figure 5. Since submission we have experimented with automated toolboxes from http://uemweb.biomed.cas.cz/tpp/features.html and http://www.columbia.edu/cu/biology/faculty/yuste/Methods/Caltracer2_5.zip but these have been unable to achieve competitive results in our hands.

As Rev_2 suggests, we do indeed intend to make the code freely available as a downloadable toolbox; indeed, it is already being used in a number of laboratories.

In response to Rev_2’s further suggestions: we have acquired more experience with data from other stains/markers and other types of microscopes and so will be able to exploit many different data types in the toolbox. Several new options the user might wish to control are also included and we are currently working on a 3D implementation of the algorithm as several of our collaborators have expressed interest in it. In cases where temporal information may be useful, we added independent components analysis (ICA) to the toolbox as a post-processing step: for each cell shape recognized by the algorithm, we run ICA in its very local neighborhood to determine precise ROI filters and exclude pixels from superimposed cells. So far we had mixed results: cells with large functional signals are indeed easily segmented by ICA localized to the cell location, but cells with poor SNR are not. We think in general the success of this step will be highly dependent on the Calcium marker used and other experimental conditions. We note that temporal information may also be used in a pre-processing step of creating a so-called correlation map and running our algorithm on the map: http://labrigger.com/blog/2013/06/13/local-cross-corr-images.

Another potential confound Rev_2 notes is that the imaging location might drift over the course of an experiment or be consistently perturbed by the animal’s movements. We do have experience with such data and found it was sufficient to align the full images with standard image registration algorithms. The remaining alignment errors are typically in the subpixel range and impact very little the shapes of the cells (>10 pixels diameter) in the mean image.

To clarify an observation Rev_4 makes in their summary, binary weights are used to determine cell location, but we also extract continuous coefficients that represent the shape of each cell in the basis set provided by the learnt subspaces. Also in response to Rev_4, larger zoom versions of figure 3 are available in the supplementary.